# Position: Machine Learning Research Should Be Guided by Explicit, Pluralistic Models of Human Purpose

**Utsav Gupta** [1]

## Abstract

Machine learning systems increasingly shape attention, work, education, and social life, yet ML research often treats the question "what is this for?" as external, relying on proxies such as accuracy, engagement, or preference satisfaction. This position paper argues that ML research should be guided by explicit, pluralistic models of human purpose, understood as supporting people's capacity to pursue meaningful, self-chosen life projects with agency. The paper proposes three community practices: (i) purpose articulation, a structured "Purpose Statement" that specifies intended beneficiaries, mechanisms, and falsifiable failure modes; (ii) purpose evaluation, which measures impacts on agency and meaning alongside task performance and harm; and (iii) purpose governance, which updates purpose frameworks through transparent, participatory processes to reduce unaccountable value-setting. This framing enables concrete technical research directions, including objective design beyond preference satisfaction, benchmarks for agency and meaning, pluralistic system behavior, and institution-aware alignment. The paper provides stakeholder-differentiated recommendations for researchers, benchmark creators, conference organizers, and funders, and addresses credible objections including value neutrality, feasibility and measurement validity, the claim that harm prevention is sufficient, and risks of ideological capture or paternalism.

## 1. Introduction

Machine learning (ML) has become a general-purpose infrastructure that shapes what people attend to, what they can do, what they believe is possible, and what kinds of

work and relationships are available to them. Consider an AI tutoring system that maximizes session length as a proxy for learning: students may spend more time on the platform while developing less capacity for independent study. Or consider a content recommendation system that optimizes engagement: users receive more of what holds their attention, but may find their curiosity narrowed rather than expanded. These are not bugs. They are natural consequences of a technical culture that treats questions of "what is this for?" as external to research—to be resolved by users, markets, or policymakers rather than by the researchers who design objectives, curate data, and choose evaluation metrics.

**Position:** *ML research should be guided by explicit, pluralistic models of human purpose, and the community should develop norms for reporting and evaluating how research contributes to (or undermines) people's ability to pursue meaningful, self-chosen life projects.*

Rather than imposing one philosophical doctrine, this position acknowledges a practical reality: ML research already encodes normative assumptions through how it formulates problems—through objectives, datasets, evaluation metrics, and deployment contexts (Crawford, 2021; Bender et al., 2021; Thomas & Uminsky, 2020; Passi & Barocas, 2019). If those assumptions are not made explicit, the community silently defaults to whatever proxies are convenient to optimize—accuracy, engagement, preference satisfaction, cost reduction—even when those proxies conflict with what people ultimately care about. The result is a field that is technically sophisticated but normatively inarticulate: capable of building systems that reshape human life, yet lacking the vocabulary and practices to ask whether those systems serve human purposes or undermine them.

**Scope: a thesis and a mechanism.** Two levels of the argument are worth separating. The *thesis*—that ML research embeds normative assumptions, better made explicit than left implicit—applies broadly. The *practical mechanism* we propose to act on it (the Purpose Statement of Section 4.1) is narrower: it is scoped to work claiming real-world relevance—papers discussing deployment, user studies on realistic tasks, or systems intended for public use. Foundational capability research (e.g., optimization, archi-

[1]Stanford University, Stanford, CA, USA. Correspondence to: Utsav Gupta <vastu@stanford.edu>.

*Proceedings of the 43rd International Conference on Machine Learning*, Seoul, South Korea. PMLR 306, 2026. Copyright 2026 by the author(s).

tectures, training stability) is not the mechanism's primary target, even though the thesis still describes it. Keeping the two levels distinct lets the position be ambitious without becoming a universal review burden.

Recent ICML position work has already pushed the community toward wider framings: pluralistic alignment (Sorensen et al., 2024), democratic governance of alignment (Ovadya et al., 2025), and "full-stack" alignment that includes institutions and flourishing (Edelman et al., 2025). The contribution of this paper is to argue that *human purpose* is a missing organizing concept that can unify these directions and make them operational for mainstream ML research. Where pluralistic alignment asks *whose values?*, and democratic alignment asks *who decides?*, purpose-aware ML asks *what is this for?*—and demands that the answer be explicit, testable, and revisable.

The remainder of the paper defines "human purpose" in a research-relevant way (Section 2); argues that purpose is an ML problem, not merely a policy concern (Section 3); proposes three community practices that operationalize purpose without requiring a single moral doctrine (Section 4); identifies five concrete research directions (Section 5); offers stakeholder-differentiated recommendations (Section 6); and addresses credible objections (Section 7).

## 2. What Is Meant by "Human Purpose"

"Purpose" can sound metaphysical. This paper uses a practical definition that is both pluralistic and research-relevant:

> **Human purpose:** the capability of people and communities to pursue meaningful, self-chosen life projects, in ways that preserve agency, dignity, and room for moral and cultural diversity.

This definition deliberately avoids a single "correct" purpose. Instead, it emphasizes three dimensions:

**Agency.** People can form goals, revise them, and act on them without manipulation or coercion. Agency requires not only that options exist but that people have the cognitive and social conditions to exercise genuine choice—what self-determination theory calls autonomy, competence, and relatedness (Deci & Ryan, 2000). Psychological research on human agency emphasizes the capacity for intentionality, forethought, self-regulation, and self-reflection as core agentic properties (Bandura, 2006), while Sen's account of agency freedom highlights the importance of people's ability to act on behalf of goals they value (Sen, 1985).

**Meaning.** People can connect actions to reasons they endorse (individually and socially), not only to transient preferences. Meaning involves what Wolf (2010) characterizes as active engagement with projects of objective worth—a

standard that goes beyond hedonic satisfaction and beyond preference fulfillment. Validated psychological instruments already measure both the presence of and search for meaning in life (Steger et al., 2006).

**Pluralism.** Different people and cultures endorse different ultimate ends; ML systems should not collapse that diversity into a single objective by default. Cross-cultural research on basic human values demonstrates both universal structure and substantial individual and cultural variation (Schwartz, 2012). The philosophical tradition of value pluralism holds that genuinely distinct and sometimes incompatible values can each be objectively valid (Berlin, 1958). Purpose-aware ML does not require convergence on a single value hierarchy; it requires that systems respect and support this diversity.

**Relationship to existing frameworks.** Several traditions sit nearby. Flourishing and well-being frameworks include "meaning and purpose" as a core domain (VanderWeele, 2017; Ryff, 1989), and recent AI work has begun to benchmark models against such constructs (Hilliard et al., 2025). The capabilities approach grounds development and justice in people's substantive freedoms (Sen, 1985; Nussbaum, 2003); Value Sensitive Design provides a methodology for integrating values at the system-design level (Friedman et al., 2013). Purpose as we define it draws on all of these but serves a different function: it is a *community-wide research norm* for ML practitioners, not a philosophical account of the good life, a design methodology, or a policy framework. The capabilities approach tells us which freedoms matter but not how researchers should report, evaluate, or govern systems' effects on them; VSD operates at the team–stakeholder level rather than the level of shared research practice; and flourishing frameworks identify the dimensions of a good life but have not yet been translated into ML evaluation infrastructure. Purpose-aware ML asks what ML research *practices* should look like given those traditions, and the three proposals of Section 4 are the answer.

**Tensions between traditions.** These traditions are not mutually consistent, and we do not claim otherwise. The capabilities approach is itself internally contested: Nussbaum defends a specified list of central capabilities; Sen declines to fix one, holding that the relevant capabilities should be settled through public reasoning in context (Nussbaum, 2003; Sen, 1985). Self-determination theory posits universal psychological needs, while value pluralism emphasizes the irreducible plurality of ends (Deci & Ryan, 2000; Berlin, 1958). Rather than adjudicate, the definition above is pitched at the level of *structural conditions* for purposeful living—the capacity to choose, find meaning, and differ—rather than the *content* of a good life. This is the level at which the Sen–Nussbaum disagreement can be bracketed rather than resolved: a research norm can ask whether a system pre-

serves people's capacity to choose and revise ends without picking a canonical list. We use *agency*, *meaning*, and *pluralism* in those structural senses throughout.

**Purpose is not preference.** A crucial distinction separates purpose from preference. Preferences can be manipulated, manufactured, or disconnected from what people reflectively care about (Susser et al., 2019). A user may prefer to continue scrolling a social media feed while reflectively endorsing a goal of spending less time on screens. Classic work on adaptive preferences shows that people's expressed wants can be shaped by the very constraints they face (Elster, 1983), and Nozick's experience machine illustrates that people care about more than experiential satisfaction—they want to *do* and *be*, not merely feel (Nozick, 1974). Purpose-aware ML takes this gap seriously: the question is not "what does the user want right now?" but "does this interaction support the user's capacity for self-directed, meaningful action over time?"

**Purpose is not well-being.** Purpose differs from subjective well-being: well-being frameworks measure how people *feel*; purpose frameworks ask whether people can *do* what they have reason to value (Ryan & Deci, 2001). A person can report high satisfaction while their agency is being eroded, and the pursuit of meaningful projects often involves difficulty that reduces momentary well-being. Systems that optimize for user satisfaction may systematically undermine purpose.

## 3. Why Purpose Is an ML Problem

Three arguments establish that purpose is not merely a policy concern but a core ML research problem.

### 3.1. ML Systems Optimize Proxies, and Proxies Reshape People

ML research typically optimizes what is easy to measure. That makes sense scientifically, but when ML becomes infrastructure, optimizing proxies can become a societal steering mechanism:

- **Optimizing engagement** can reward attention capture rather than understanding or self-directed goals, as the attention economy and surveillance capitalism illustrate (Williams, 2018; Zuboff, 2015).

- **Optimizing productivity** can displace meaningful work or deskill workers, particularly in tasks that provide skill development and occupational meaning (Hazra et al., 2025; Acemoglu, 2024; Klinova & Korinek, 2021).

- **Optimizing preference satisfaction** can amplify short-term desires and learned dependencies, undermining

the deliberative processes through which people form and revise deeper goals.

The core issue is not that proxies are "bad." It is that over-reliance on metrics invites predictable failure: when a measure becomes a target, it ceases to be a good measure, and optimization pressure leads to gaming, short-termism, and displacement of the outcomes metrics were meant to capture (Thomas & Uminsky, 2020). Proxies become de facto definitions of success unless the community explicitly contests and corrects them. When foundation models are deployed at scale (Bommasani et al., 2021), the gap between proxy optimization and purpose-relevant outcomes becomes a systemic concern rather than an edge case.

### 3.2. "Alignment" without Purpose Is Underspecified

Current alignment practice often aims at "following instructions," "helpful and harmless behavior," or "matching user preferences" (Ouyang et al., 2022; Bai et al., 2022). These are useful, but incomplete:

- Instructions can be inconsistent, manipulated, or harmful to third parties—faithful instruction-following does not imply alignment with users' deeper interests (Gabriel, 2020).

- "Harmlessness" does not capture positive human goods; a system can be perfectly harmless and utterly useless for human flourishing (Green, 2019).

- Preference satisfaction struggles with addictive or socially constructed preferences and cannot distinguish endorsed values from momentary choices (Gabriel, 2020).

Recent work arguing for thicker models of value and full-stack alignment highlights related failures (Edelman et al., 2025): intent-aligned systems embedded in misaligned institutions can still produce anti-flourishing outcomes. Purpose-aware ML extends this logic—systems can be locally aligned to a proxy while globally misaligned with people's capacities to live meaningful lives. The concrete problems in AI safety identified by Amodei et al. (2016) remain relevant, but they focus on avoiding failures rather than specifying what success looks like from the standpoint of human purpose. Russell (2021) argues that machines should be uncertain about human preferences and defer to humans accordingly, yet uncertainty alone does not resolve the deeper problem: preferences themselves can be manipulated, manufactured, or disconnected from reflective endorsement.

### 3.3. The Gap Between Safety and Purpose

A system can be safe—avoiding catastrophic outcomes, respecting constraints, declining harmful requests—while still being *anti-purpose*: systematically eroding the conditions under which people can pursue meaningful lives. This gap deserves explicit attention.

Consider an example. A language model that consistently provides correct, harmless answers to homework questions may simultaneously undermine students' development of independent reasoning—the very capacity the educational context is meant to build. The system passes conventional safety evaluations; the harm is not in what it does wrong, but in what it displaces: agency, skill development, and the effortful engagement through which people construct meaning. The choice to optimize for "correct answers" rather than, say, for productive struggle is a normative commitment made at the *research* phase—when the objective and evaluation metric are selected—not only at deployment. The proxy that will later reshape students is fixed the moment the objective is chosen, which is why the gap between safety and purpose is a research problem and not merely a downstream policy concern.

This pattern is not hypothetical: technology-induced deskilling has been documented across domains (Danaher & Nyholm, 2021; Bankins & Formosa, 2023), and as ML systems grow more capable it expands to higher-order capacities (planning, deliberation, professional judgment). Risk taxonomies and ethical frameworks have noted human flourishing as distinct from harm avoidance (Weidinger et al., 2021; Floridi et al., 2018), but those recognitions have not yet translated into standard ML research practices.

The structural reason for this gap is that safety is a *constraint* while purpose is an *objective*. Safety asks "does this system avoid bad outcomes?"—a question that can be evaluated with bounded test cases. Purpose asks "does this system contribute to good outcomes for human lives?"—a question that requires richer evaluation, longer time horizons, and engagement with what "good" means in context. The purpose-aware orientation proposed here aims to close that gap by giving the community tools to ask and answer the second question alongside the first.

## 4. A Purpose-Aware ML Agenda

This paper proposes three community practices that would make "purpose" actionable without requiring a single moral doctrine. Table 1 summarizes the shift from current norms to purpose-aware norms across several dimensions of ML research practice.

**Conflicts are the point, not a bug.** Purpose dimensions can pull against each other—agency may call for offering more choices while meaningfulness may call for more focused guidance—and different stakeholders' purposes can conflict outright, as when a teacher's interest in students' long-run autonomy is in tension with students' interest in short-term efficiency. The proposal does not pretend these tensions away; making them *visible* is precisely its value. The three practices divide this labor: articulation forces a system's designers to name which dimension takes priority in which context (e.g., guided meaning-making for novices, autonomous exploration as competence develops) and to declare whose purposes count via the "Whose purposes?" component; evaluation tests whether the stated priority actually holds in use; and governance (Section 4.3) supplies the mechanism for outside parties to contest and revise those priority choices over time. Disclosure alone would be too weak; it is the combination of articulation, evaluation, and governance that turns an unavoidable value conflict into something the community can argue about openly.

### 4.1. Purpose Articulation: A "Purpose Statement" Norm

Each ML paper that claims real-world relevance should include a short, testable **Purpose Statement** that answers:

1. **Whose purposes?** Which stakeholders are intended beneficiaries? Who bears risk?

2. **What purpose domain?** Education, health, creativity, relationships, civic life, work, scientific discovery, etc.

3. **What mechanism?** How does the method support the stated purpose?

4. **What failure modes could undermine purpose?** Manipulation, dependency, disempowerment, exclusion, deskilling, etc.

5. **What evidence would change your mind?** What observation would indicate the method is not supporting the stated purpose?

This is not ethics prose. It is a research commitment that can be evaluated, challenged, and improved—analogous to how datasheets (Gebru et al., 2021) and model cards (Mitchell et al., 2019) have made data and model documentation into accountable research artifacts. Value Sensitive Design (Friedman et al., 2013) provides a complementary methodology for integrating human values into technology design from the outset. The Purpose Statement extends this documentation norm from describing *what a system is* to specifying *what a system is for*. Concretely, it is meant to *complement* rather than replace existing documentation: a Purpose Statement can live as a dedicated section within a model card or as a standalone artifact linked to one. Where a model card reports architecture, training data, and measured

*Table 1.* Current ML research norms compared with purpose-aware norms across key dimensions of research practice.

| Dimension | Current Practice | Purpose-Aware Practice |
| --- | --- | --- |
| Objective | Task accuracy, reward, preference match | Includes agency preservation, meaning support, pluralism |
| Evaluation | Benchmarks for performance and harm | Adds measures of agency, meaning, dependency, deskilling |
| Documentation | Model cards, datasheets (what the system *is*) | Purpose Statement: what the system is *for*, for *whom*, with what failure modes |
| Success criterion | Higher score on proxy metric | Proxy metric *plus* evidence of purpose-relevant impact |
| Alignment target | Follow instructions; be helpful and harmless | Support reflective self-direction; complement human capacities |
| Governance | Lab-internal, market-driven, or regulatory | Participatory, transparent, revisable purpose frameworks |

performance, the Purpose Statement records the human end those measurements are standing in for.

**Which papers need one.** Consistent with the scope distinction drawn in Section 1, the norm targets work claiming real-world relevance, not foundational capability research; a paper on transformer training stability would not need a Purpose Statement, whereas a paper reporting medical diagnosis accuracy would benefit from stating *whose* diagnostic purposes it serves and which failure modes (e.g., clinician over-reliance, deskilling of diagnostic judgment) it anticipates. The test is not whether the work is "applied" but whether "accuracy" or "helpfulness" is silently standing in for a human purpose that deserves to be named.

To illustrate, consider a hypothetical Purpose Statement for an AI writing assistant: "*This system is intended to support professional writers (beneficiaries) in creative work (domain) by generating drafts that the writer revises and controls (mechanism). Failure modes include dependency (the writer stops generating original ideas), deskilling (the writer's independent writing ability declines), and homogenization (outputs converge on a narrow stylistic range). Evidence that would change our assessment includes longitudinal studies showing reduced writer output diversity or self-reported loss of creative agency.*" A Purpose Statement of this kind makes normative commitments visible, testable, and open to challenge.

Purpose Statements vary in quality. A *weak* statement offers boilerplate: "This system helps users by providing relevant information." An *adequate* statement specifies components: "This system supports first-generation college students (beneficiaries) in course selection (domain) by surfacing peer outcomes and advisor recommendations (mechanism). Key failure mode: over-reliance leading to reduced self-efficacy in academic planning." A *strong* statement adds falsifiable criteria: "We would revise if longitudinal surveys show decreased confidence in making decisions without system assistance." The quality gradient is deliberate: it lets the norm be adopted incrementally rather than imposing a uniform

testing burden, and it is meant to be *generative*—surfacing research questions that current norms leave implicit—rather than a uniform standard every paper must meet (we return to the feasibility of testing such claims in Section 7, Alternative View D).

**Not a product mission statement.** A Purpose Statement should not be confused with a corporate mission statement, which is aspirational, internally facing, and not subject to external verification. A Purpose Statement differs structurally in three ways: (i) it is organized around specific components (beneficiaries, mechanisms, failure modes, falsification criteria) rather than aspiration; (ii) it is public and contestable—subject to peer review and external audit via the registries of Section 4.3; and (iii) it commits in advance to failure modes and to evidence that would count against it, which mission statements never do. The closest analogue is a clinical trial protocol's pre-registered statement of intended outcomes and stopping rules, not a brand promise.

**Who writes it, and accountability.** Who authors a Purpose Statement materially affects what it is worth: a statement written by a lab about its own system is a weaker accountability mechanism than one co-developed with affected communities. We therefore propose a *tiered* approach. For research papers, the authors draft the statement, with peer review serving as the initial check—this is imperfect but still creates a falsifiable artifact that auditors and affected communities can later contest, where previously there was nothing concrete to push back against. For deployed systems, where stakes and power asymmetries are higher, Purpose Statements should incorporate participatory input from the people who bear the risks, drawing on the governance mechanisms of Section 4.3. The lab-authored statement is thus necessary but not sufficient: it is the entry condition for accountability, not its endpoint.

**When triggering evidence arrives.** The "what evidence would change your mind?" clause is only meaningful if something follows when that evidence appears. We envision a *graduated* response rather than a single binary

verdict. Minor or early-warning signals warrant design iteration—for example, adjusting defaults to reduce a dependency effect. Serious signals—say, longitudinal evidence of systematic deskilling—warrant escalation through the community-review and open-registry mechanisms of Section 4.3, prompting external scrutiny and, where warranted, withdrawal or redesign. The model is clinical-trial adverse-event reporting: predefined triggers, proportionate escalation, and a public record, not a one-time pass/fail gate.

## 4.2. Purpose Evaluation: Measuring Agency and Meaning

The community needs evaluation approaches that capture purpose-relevant outcomes. Depending on the application, these can include:

**Agency measures.** Does the system preserve user control? Does it increase option value and reversibility? Does it reduce lock-in or coercive dependence? Agency evaluation could draw on constructs from self-determination theory (Deci & Ryan, 2000)—measuring perceived autonomy, competence, and relatedness before and after system interaction.

**Meaning measures.** Does the system help users clarify goals, understand tradeoffs, and act on endorsed reasons—or does it substitute for reflection? Validated instruments such as the Meaning in Life Questionnaire (Steger et al., 2006) and short flourishing scales (Diener et al., 2010) provide starting points, though adaptation for system-interaction contexts is needed. The IEEE 7010 standard for assessing AI impact on human well-being (IEEE, 2020) offers a procedural template for such adaptation.

**Pluralism measures.** Does the system appropriately represent a range of legitimate perspectives, or does it collapse toward a narrow style, ideology, or persona? Frameworks such as ValueCompass (Shen et al., 2024) and the PRISM project (Kirk et al., 2024) offer structured approaches to mapping value diversity and assessing whether systems respect it. Measuring pluralism is technically challenging because it requires distinguishing meaningful diversity from noise; purpose-aware pluralism evaluation remains an open research area (see Direction 5).

Purpose evaluation will often require mixed methods: controlled user studies where feasible; longitudinal evaluation to detect dependency or deskilling; benchmarks designed around flourishing and meaning constructs (Hilliard et al., 2025); and audits focused on manipulation, autonomy erosion, and stakeholder harms. Holistic evaluation frameworks (Liang et al., 2022) provide a precedent for multidimensional assessment, though they have not yet incorporated purpose-relevant dimensions.

Several practical challenges must be acknowledged:

purpose-relevant outcomes often manifest over longer time horizons than standard evaluations capture; purpose is partially subjective (two users may reasonably disagree about whether a system supports their agency); and purpose evaluation requires engagement with affected populations, not only annotators. These challenges are real but not unique—similar issues arise in evaluating educational technology, public health interventions, and workplace systems. A further consequence is that purpose claims are inherently *conditional on deployment context* rather than global verdicts on a method: the same model may support agency in one setting and erode it in another. This is by design. The failure-mode and falsification components of a Purpose Statement are precisely what make a claim context-specific—and therefore testable—rather than a blanket positive or negative judgment about a technique.

**Reflexive measurement commitment.** Any purpose metric risks Goodhart failure if optimized directly. Purpose-aware evaluation requires second-order monitoring: Are researchers gaming the metric? Does the measure still predict intended outcomes? The community should treat purpose metrics as hypotheses to validate, not targets to maximize.

The key shift is that "human purpose" becomes a target for evaluation design, not a post-hoc concern.

## 4.3. Purpose Governance

A predictable objection is that "purpose is political." That is exactly why it needs governance. Without transparent, participatory approaches, the default will be: private, unaccountable value-setting by labs; implicit value-setting by datasets and metrics; or ideological capture via whichever goals are easiest to optimize. A "society-in-the-loop" approach (Rahwan, 2018) that embeds democratic feedback into algorithmic governance offers one path forward. The proliferation of AI ethics guidelines globally (Jobin et al., 2019) demonstrates demand for governance, but principles alone cannot guarantee ethical outcomes without institutional mechanisms to enforce them (Mittelstadt, 2019).

Pluralistic alignment (Sorensen et al., 2024) and democratic alignment (Ovadya et al., 2025) proposals offer building blocks: purpose frameworks can be defined at different levels (organizational, sectoral, national, global), with increasing degrees of participation and contestability. The goal is not perfect consensus; it is legitimate, revisable decision-making. The sociotechnical perspective that fairness cannot be meaningfully defined in abstraction from institutional context (Selbst et al., 2019) applies equally to purpose: purpose frameworks must be situated, not universal.

Purpose governance also requires institutional support. Possible mechanisms include:

- **Open registries** of Purpose Statements (analogous to clinical trial registries) that allow external audit and longitudinal tracking of whether stated purposes are actually evaluated.

- **Community review processes** that periodically assess whether purpose claims are being tested and whether evaluation results lead to design changes.

- **Cross-disciplinary forums**—conference tracks, workshops, or standing committees—where purpose claims are debated across ML, social science, philosophy, and affected communities.

- **Participatory input mechanisms** that give affected populations a voice in defining what "purpose" means in specific deployment contexts, drawing on democratic alignment proposals (Ovadya et al., 2025).

These mechanisms create infrastructure for legitimate contestation—a process by which the community can ask "whose purposes are being served?" and receive a verifiable answer. Without such infrastructure, purpose discourse risks becoming performative.

### 4.4. A Worked Example: From Statement to Governance

To show the three practices operating end to end, we follow a single system—an AI writing assistant for professional writers—through all of them.

**Articulation and evaluation.** The team publishes the Purpose Statement sketched in Section 4.1: beneficiaries are professional writers; the mechanism is draft generation that the writer revises and controls; the declared failure modes are dependency, deskilling, and homogenization; and the falsification clause commits to revisiting the design if longitudinal data show reduced output diversity or self-reported loss of creative agency. These commitments dictate what to measure beyond the task metric (e.g., draft acceptance rate). Adapting the agency protocol of Section 5 (Direction 2), the team tracks the lexical and structural diversity of a writer's output over months (homogenization), the share of finished text accepted verbatim from the model versus written by the author (dependency), and a periodic self-report instrument on creative agency—paired with an anti-Goodhart check that credits appropriate divergence from suggestions rather than mere agreement.

**Governance, and what the framework adds.** Suppose the longitudinal data show acceptance rates *rising* while output diversity falls and self-reported agency declines—precisely the falsification condition. Under the graduated response of Section 4.1, the finding is logged to the open registry (Section 4.3), triggers community review, and prompts a design change (e.g., surfacing several stylistic options by default, or eliciting an original opening before generation). A product mission statement ("empower writers") would have been satisfied by the rising acceptance rate alone; the Purpose Statement is not. By naming dependency and homogenization as failure modes *in advance* and committing to evidence that would count against the design, it reframes a proxy win as a recognized purpose failure, and the governance layer ensures someone is obliged to act on it. That is the insight mission-statement thinking structurally cannot produce.

## 5. Research Directions

A purpose-aware orientation suggests concrete technical and scientific work. The following directions are not exhaustive, but each is tractable with existing methods and addresses a gap in current research.

**Direction 1: Objective design beyond preference satisfaction.** Methods that distinguish reflective endorsement from impulsive preference, and that model long-term agency and meaning impacts. This goes beyond current RLHF paradigms by asking not only "what does the user want right now?" but "does this interaction support the user's capacity for self-directed action over time?" Technically, this may involve multi-objective optimization with agency-relevant auxiliary objectives, reward modeling that incorporates temporal discounting of purpose-relevant outcomes, or training procedures that weight reflective feedback (e.g., post-session evaluations) differently from in-the-moment signals. Existing work on constitutional AI (Bai et al., 2022) already separates training signals from direct user preference; purpose-aware objective design extends this separation by grounding the additional signal in agency and meaning constructs.

**Direction 2: Benchmarks for agency and meaning.** Evaluation suites covering goal formation, goal revision, tradeoff reasoning, and resistance to manipulative optimization. A human-centered evaluation paradigm (Lindauer et al., 2024) can be extended to purpose-relevant constructs. Concretely, an agency-evaluation protocol for a decision-support system would measure pre-interaction goal clarity, rationale quality, and confidence on a realistic open-ended task (e.g., course selection, career planning); have participants use the system for a recommendation-assisted decision; then assess goal ownership (do they articulate the goal as their own?), rationale quality (can they explain the decision in their own terms?), and critical updating (when given new information the system did not consider, do they update or defer?). An anti-Goodhart check is essential: the system passes if users deviate appropriately from recommendations when warranted, not if they maximize agreement.

**Direction 3: Pluralistic system behavior.** Systems that can represent multiple legitimate viewpoints and avoid "one-size-fits-all" alignment. This connects to existing work on pluralistic alignment (Sorensen et al., 2024) but adds an explicit requirement that pluralism serve human agency rather than merely reflecting statistical diversity. A concrete, tractable contribution would be a *steering mechanism* that lets users specify the value framework a system should operate within—for example, different cultural norms around directness in advice-giving, or differing preferences for autonomy versus guidance in a learning tool—together with an evaluation that tests whether the system's outputs actually shift in response to those controls. This extends the descriptive diversity measurement of the PRISM methodology (Kirk et al., 2024) toward *controllable* pluralism, and it is sharper than simply training on more diverse data: it requires explicit user-facing controls and a demonstration that the controls are effective. The challenge is to support genuine diversity—including minority and non-Western conceptions of purpose—without collapsing into relativism or enabling harmful uses (Schwartz, 2012).

**Direction 4: Institution-aware alignment.** Methods that model how deployment contexts—platform incentives, labor markets, governance structures—interact with model behavior to produce purpose-relevant or purpose-undermining outcomes (Edelman et al., 2025). A system that is purpose-aligned in isolation may become purpose-undermining when deployed within an institution whose incentives conflict with user flourishing. For example, an AI writing tool that supports creative agency in an individual context may deskill and displace workers when deployed as a cost-reduction tool in a corporate context with misaligned incentives. A concrete, tractable contribution here is an *evaluation benchmark that parameterizes institutional context*: a suite of simulated deployment scenarios—e.g., the same AI tutoring system embedded under a school-aligned incentive structure versus an engagement-optimizing one—against which researchers can test whether a system's purpose-relevant outcomes degrade. This is what distinguishes the contribution from a policy analysis: it is reusable ML evaluation infrastructure, an artifact that lets researchers stress-test systems against varied institutional conditions, exactly analogous to how robustness benchmarks parameterize distribution shift. Complementary work could include causal models of system–institution interaction and methods for detecting when deployment conditions undermine a system's stated purpose.

**Direction 5: Measurement validity for purpose constructs.** Meta-research on whether "purpose metrics" predict real outcomes, and when they fail. The community should adopt an empirical stance toward its own normative instruments: propose measures, test predictive validity, revise, and openly document uncertainty. Specific research questions include: Do proxy measures of agency (e.g., user control preservation) predict long-term outcomes like skill maintenance? Do meaning measures (e.g., self-reported purpose satisfaction) correlate with behavioral indicators of engagement quality? What are the failure modes of purpose metrics—when do they systematically misrepresent purpose-relevant outcomes? This direction is foundational: without validated measurement, purpose-aware ML risks becoming aspirational rhetoric rather than research practice.

## 6. Call to Action

**For ML researchers.** Include a Purpose Statement in papers claiming real-world relevance (work discussing deployment, user studies on realistic tasks, or systems intended for public use). Even a brief two-to-three-sentence statement specifying intended beneficiaries, purpose domain, and key failure modes is a meaningful advance over current practice. Treat "purpose failure modes" as first-class in ablations and evaluation, on par with robustness or fairness, and report evidence on whether the method supports or undermines agency, meaning, and pluralism.

**For dataset and benchmark creators.** Build benchmarks that test agency, meaning, and pluralism, not only task accuracy and harm avoidance; existing flourishing instruments from psychology and public health offer a starting point (VanderWeele, 2017; Steger et al., 2006). Publish documentation describing what conception of "good outcome" the benchmark assumes, including explicit acknowledgment of which purposes are *not* captured.

**For ICML and other conference organizers.** Encourage (or pilot) a lightweight structured field in submission forms—"Intended human purpose and evaluation plan"—and incentivize replication and longitudinal evaluation where purpose impacts are plausible.

**For funders and labs.** Fund cross-disciplinary teams (ML + social science + philosophy + HCI) to build validated purpose evaluation instruments; the gap between existing psychological instruments and ML evaluation needs is a concrete research opportunity. Reward open evaluations and external audits that assess purpose-relevant outcomes alongside standard performance metrics. Publish longitudinal impact assessments of deployed systems that track purpose-relevant outcomes (agency, skill, meaning) over time, not only at the point of deployment.

**A note on incrementalism and cost.** These recommendations are deliberately incremental: they ask for *transparency* (state what the system is for), *evaluation* (measure whether it achieves that purpose), and *accountability* (let others challenge and revise the claim), extending existing norms (datasheets, model cards, impact statements) rather than replacing them. The *minimal viable* adoption is a two-

to-three-sentence Purpose Statement naming beneficiaries, domain, and key failure modes—no interdisciplinary team or new budget required. The heavier practices (longitudinal evaluation, validated instruments, registries) are proposed as shared *community infrastructure*, built once and reused, not a per-paper tax. Purpose is an *additional* objective alongside safety; quantifying the costs of different alignment failures is itself a problem Direction 5 (Section 5) takes up.

**A note on communication.** The Purpose Statement's structured format—beneficiaries, mechanisms, failure modes, falsification criteria—lends itself to audience-specific renderings: the same commitments can be surfaced for policymakers (who bears which risks), practitioners (which failure modes to monitor in deployment), and the public (what the system is for, and how one would know if it stopped serving that purpose). Building such layered guidelines is valuable follow-up work, not a prerequisite for adoption.

# 7. Alternative Views

**Alternative View A: "ML should be value-neutral; purpose is for users and policymakers."**

*Claim:* Science should produce capabilities, not decide ends. Embedding purpose in research risks politicizing ML.

*Response:* The politicization concern is genuine—any framework that makes normative commitments visible creates a surface for contestation, and purpose discourse can be co-opted by institutional, ideological, or commercial agendas. But ML is not value-neutral in practice: objectives, data, and metrics already encode values (Crawford, 2021), and artifacts have politics regardless of designer intent (Winner, 1980). The question is whether values enter visibly or invisibly. Pretending neutrality hides them and shifts them to whoever controls deployment or incentives. Making value assumptions explicit and testable is closer to scientific transparency, not further from it. The structural mitigations against capture are that purpose frameworks be pluralistic (no doctrine privileged), revisable (open to challenge), and transparent (stated publicly, not hidden in design choices). The alternative—implicit value-setting through metric choice—is not less political, only less accountable.

**Alternative View B: "Purpose is subjective and culturally contested; any framework will impose ideology."**

*Claim:* "Human purpose" varies by culture and person. Formalizing it invites ideological capture.

*Response:* The move is to treat purpose as a pluralistic, governed object. The definition in Section 2—agency, meaning, pluralism—specifies *conditions* for purposeful living rather than *content*. Cross-cultural research finds that specific values differ but their structure has substantial regularity (Schwartz, 2012). Capture is more likely when values are

implicit than when visible and contestable.

**Alternative View C: "Harm prevention and rights compliance are sufficient; purpose is extra."**

*Claim:* It is enough to prevent harm and follow safety norms (Amodei et al., 2016). Purpose is beyond ML's scope.

*Response:* Harm prevention is necessary but not sufficient: disempowerment, dependency, and deskilling can occur even when systems are "safe." As argued in Section 3.3, a system can pass every safety evaluation while eroding the conditions under which people construct meaning.

**Alternative View D: "This is infeasible. ML cannot measure meaning or purpose reliably."**

*Claim:* Purpose constructs are too noisy and subjective to be engineering targets.

*Response:* Existing instruments for meaning and agency were built for self-report in clinical or survey contexts, not ML interactions, and long-horizon outcomes (deskilling, erosion of intrinsic motivation) may resist short-horizon measurement. The proposal does not claim current instruments are deployment-ready; it argues that building validated purpose measures is a tractable, necessary program. Psychology, public health, and social science already build imperfect-but-valid instruments (Steger et al., 2006; Ryff, 1989; Deci & Ryan, 2000), and ML can adopt the same empirical stance. The same "too hard to measure" was once said of fairness; Direction 5 (Section 5) treats measurement validity itself as a research priority.

# 8. Conclusion

ML increasingly shapes the conditions under which people decide what to do with their lives. If ML research continues to optimize narrow proxies without explicit purpose commitments, the community risks building systems that are highly capable yet systematically anti-purpose: eroding agency, narrowing life options, and displacing meaning.

The deepest risk is not that ML systems will cause dramatic harms—the safety community is addressing that. The risk is that ML systems will quietly reshape human life in ways that no one explicitly chose and no one can easily contest: making people more dependent, less skilled, less agentic, and less able to articulate what they are living for. Purpose-aware ML is the community's opportunity to ensure that what it builds is not only powerful and safe, but worth building.

# Impact Statement

This paper proposes changes to ML research norms and evaluation practices, arguing that the community should make

human purpose an explicit, first-class concern. The intended broader impact is to encourage researchers, benchmark creators, conference organizers, and funders to adopt practices that foreground agency, meaning, and pluralism alongside standard performance metrics. A risk is that any formalization of "purpose" could be co-opted to impose particular value systems; the paper addresses this risk by advocating pluralistic, participatory governance of purpose frameworks. The paper does not introduce new models, datasets, or algorithms, and its direct societal impact is mediated through the adoption (or non-adoption) of the proposed norms by the research community.

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
