# OpenReview forum: "Position: Machine Learning Research Should Be Guided by Explicit, Pluralistic Models of Human Purpose"
_ICML.cc/2026/Position_Paper_Track — ICML 2026 Position Paper Track regular_

### Official Review · Reviewer_EWSy · 2026-03-13

**Significance:** 3
**Argument Clarity:** 2
**Rating:** 5
**Confidence:** 4

**Questions:**

See weaknesses.

**Alternative Views Section:**

Yes

**Compliance With Llm Reviewing Policy A Conservative:**

Affirmed.

**Discussion Potential:**

3

**Final Justification:**

The authors have addressed my concerns and convincingly, and I have updated my score accordingly. The paper proposes a very actionable change to research that attempts to tackle real world relevance.

**Paper Summary:**

The paper argues that machine learning research should be guided by / optimizing purpose, rather than just optimizing shorter-term proxies, aligning to preferences, or minimizing safety harms. The paper proposes developing the norm of a purpose statement for ML research, as well as evaluation measures for purpose, and some governance structure (e.g., society in the loop). Research directions include designing objectives and benchmarks, advancing pluralistic and institution aligned behavior, and performing meta research on purpose metrics for measurement validity. The paper offer us multiple calls to actions for various stakeholders and response to various alternative views including those around in feasibility, political ideology, and necessity of purpose beyond harm reduction.

**Position:**

Yes

**Position In Title:**

Yes

**Related Work:**

3

**Strengths And Weaknesses:**

Strengths:
1. The paper is well organized and well written and proposes a position that is very relevant for machine learning research.
2. The paper provides reasoning for why current practice can be insufficient, e.g. issues that arise from over optimizing for proxies.
3. The paper offers both a wide vision for carrying out research as well as incremental changes that could be implemented practically and immediately.
Weaknesses:
1. There still exist many basic capabilities with large headroom for improvement, e.g., improving the basic reliability of models. It is unclear how the exercise of writing a purpose statement, etc. would benefit this sort of research.
2. It would be useful to delineate between writing a purpose statement for a research agenda versus a product goal, as the proposed vision seems similar to the exercise of design designing mission statements and products around these.
3. The paper could benefit from a deeper dive into a case study example to highlight concrete benefits of this perspective.

**Support:**

2

---

> ### Author Rebuttal · Authors · 2026-03-31
>
> We thank the reviewer for recognizing the paper's relevance and its balance of vision with incremental changes.
>
> **Relevance to basic capability research.** Work on fundamental capabilities (optimization, architectures, reliability) is not the primary target of the Purpose Statement norm; a paper on transformer training stability would not need one. The practical mechanism targets papers claiming real-world relevance (Sections 4.1, 6), where "accuracy" or "helpfulness" are proxies for a human purpose that deserves articulation. A paper on medical diagnosis accuracy would benefit from stating whose diagnostic purposes it serves and what failure modes (e.g., over-reliance, deskilling) it anticipates. We will make this boundary crisper in the revision.
>
> **Research agenda vs. product mission statement.** A product mission statement is aspirational, internally facing, and not subject to external verification. A Purpose Statement differs in three structural ways: (1) it is organized around specific components (beneficiaries, mechanisms, failure modes, falsification criteria); (2) it is public and contestable, subject to peer review and external audit via open registries; (3) it includes failure modes and falsification criteria, which mission statements do not. The closest analogy is a clinical trial protocol's statement of intended outcomes and stopping rules, not a corporate mission. We will add this distinction.
>
> **Case study.** Section 4.1 already provides an AI-assisted writing tool example with explicit failure modes (dependency, deskilling, homogenization) and falsification criteria. In the revision, we will extend this into a full end-to-end case study: the Purpose Statement identifies professional writers as beneficiaries and dependency as a key failure mode; evaluation tracks output diversity and self-reported creative agency longitudinally; if those measures worsen, governance mechanisms trigger external review. This sketch illustrates how the framework generates insights that mission-statement-style thinking does not, specifically through failure mode analysis and falsification criteria.

---

> > ### Author Rebuttal · Reviewer_EWSy · 2026-04-03
> >
> > Thank you for your response. The authors have addressed my concerns and convincingly, and I have updated my score accordingly.

---

### Official Review · Reviewer_Ss4b · 2026-03-16

**Significance:** 3
**Argument Clarity:** 3
**Rating:** 5
**Confidence:** 4

**Questions:**

The paper proposes "clear communication to non-technical audiences". Has a hierarchical communication guideline been established? For instance, considering the different needs of policy makers, industry practitioners, and the general public, how can the core differences of aligned concepts be simplified and accurately conveyed? Are there any practical communication tools or examples that can be implemented?

**Alternative Views Section:**

Yes

**Compliance With Llm Reviewing Policy A Conservative:**

Affirmed.

**Discussion Potential:**

4

**Ethics Review Area:**

["Legal Compliance (e.g., EU AI Act, GDPR, copyright, terms of use)", "Research Integrity Issues (e.g., plagiarism, collusion rings, etc.)"]

**Final Justification:**

My concerns have been addressed. I decide to maintain my current score.

**Paper Summary:**

In recent years, ML technology has become the core infrastructure reshaping fields such as education, work, and social interaction. However, the academic and industrial communities have long focused on "quantifiable technical indicators" (such as accuracy, participation rate, and satisfaction of preferences) as the core of their research, neglecting the potential conflicts between these indicators and the ultimate goals of humanity - for instance, optimizing "learning duration" may weaken students' autonomous learning ability, and optimizing "participation rate" may exacerbate users' information cocoons. The paper points out that ML research has fallen into the predicament of "technically refined but value ambiguous", and the formulation of its research questions aligns with the current practical needs of AI ethics and sustainable development of technology, possessing significant cautionary significance and leading value.

**Position:**

Yes

**Position In Title:**

Yes

**Related Work:**

3

**Strengths And Weaknesses:**

1. The paper emphasizes the diversity of human goals, but fails to provide in-depth discussion on "how to reconcile the conflicts among different goals". For instance, "agency" might require the system to offer more choices, while "meaningfulness" might need the system to provide more focused guidance; different users' goals may conflict (such as teachers' pursuit of "improvement of students' autonomy" and students' pursuit of "short-term learning efficiency").
2. The practical suggestions proposed in the paper (such as conducting longitudinal evaluations and establishing interdisciplinary teams) may require a significant amount of additional time, manpower, and financial costs, especially for small research teams or institutions with limited resources, making their adoption quite challenging.
3. The paper points out the conflicts of different alignment ideals, but fails to quantify and weigh the potential costs of various alignment failures and the benefits of successful alignment. For example, the failure of "abstention from takeover" may lead to catastrophic risks, but the implementation cost is high and the uncertainty is great.

**Support:**

3

---

> ### Author Rebuttal · Authors · 2026-03-31
>
> We thank the reviewer for the excellent scores and practical concerns.
>
> **Conflicts among purpose dimensions.** The reviewer gives a sharp example: agency may require more choices while meaningfulness may require more focused guidance. We see this as a feature: making such tensions *visible* through the Purpose Statement is the point. A Purpose Statement for an educational system would specify which dimension takes priority in which context (e.g., guided meaning-making for novices, autonomous exploration as competence develops). Stakeholder conflicts surface in the "Whose purposes?" component. The paper's architecture goes further than disclosure alone: it is articulation, evaluation, *and governance*. Purpose Statements surface the tradeoff, and purpose governance (Section 4.3) provides the mechanism for contesting and revising those priority choices over time. We will add this discussion.
>
> **Resource burden on small teams.** The minimal viable adoption is a 2-3 sentence Purpose Statement — no interdisciplinary team required (Section 6 emphasizes this incrementalism). The more resource-intensive practices are proposed as community infrastructure, not requirements for individual papers.
>
> **Quantifying costs and communication.** On costs: safety remains a necessary constraint, and purpose is an additional objective, not a substitute for catastrophic-risk or rights-based work (the paper frames safety as a constraint and purpose as an objective in Section 3.3). Quantifying the relative costs of different alignment failures is a research challenge that Direction 5 (measurement validity) is designed to address. On communication: the Purpose Statement's structured format (beneficiaries, mechanisms, failure modes) lends itself to communication across audiences, even though the primary context is research papers. Audience-specific renderings for policymakers, practitioners, and the public remain valuable follow-up work that we will highlight in the revised Call to Action.

---

> > ### Author Rebuttal · Reviewer_Ss4b · 2026-04-03
> >
> > Thank the authors for their responses. My concerns have been addressed. I decide to maintain my current score.

---

### Official Review · Reviewer_JrsL · 2026-03-16

**Significance:** 4
**Argument Clarity:** 4
**Rating:** 5
**Confidence:** 4

**Questions:**

n/a

**Alternative Views Section:**

Yes

**Compliance With Llm Reviewing Policy A Conservative:**

Affirmed.

**Discussion Potential:**

4

**Final Justification:**

Fairly positive on this paper - I think it's a bold position and clearly argued for.

**Paper Summary:**

This paper proposes "purpose" as a key principle for ML research to uphold - that is, proposed approaches should additionally be evaluated on their ability to help humans/users live purposeful lives. The authors give definitions of purpose and argue that it is important for ML research, critically as a distinct concept from safety. They provide a "Purpose-Aware ML Agenda", touching on purpose statements, measurements, and governance as important directions for the ML community.

**Position:**

Yes

**Position In Title:**

Yes

**Related Work:**

4

**Strengths And Weaknesses:**

Strengths:
- interesting set of points that I haven't heard explicitly articulated in this way before
- think the distinction between safety and purpose is critical and useful, as is the distinction between purpose and well-being. Both advance the argument in important ways and help the reader
- bold position, clearly argued for! position paper track needs more of this
- the pluralism piece of this is important for the argument, was glad the authors included that even if it felt a little orthogonal in places. Important to note

Weaknesses:
- it seems the authors blur the distinction between ML research and ML deployment - for instance in 3.3 the homework example seems more relevant to either deployed models or very applied research. As with much values-based stuff, the application to theoretical or general work is somewhat slippery - it seems challenging to expect researchers to foresee every potentially downstream application and evaluate all impacts
- not totally clear to me if the suggestion is that purpose evaluation become a part of every paper, or just a new research direction in the field as a whole
- in general I think the notion of a "testable purpose statement" is nice but a little tricky - it seems that testing the purpose statement for paper is often challenging enough to justify a whole new line of research. Additionally, it seems that for most papers, impacts on purpose would be highly dependent on details and context of deployment and difficult to summarize as positive or negative, even conditionally so

Question:
- One example of a purpose statement in the paper discusses "Evidence that would change our assessment" - what do the authors suggest happen when such evidence is uncovered and assessments are changed?

**Support:**

4

---

> ### Author Rebuttal · Authors · 2026-03-31
>
> We thank the reviewer for the strong endorsement and for engaging so constructively with the proposal's practical implications.
>
> **What happens when evidence triggers reassessment?** The reviewer raises an important question. The Purpose Statement's "evidence that would change our assessment" clause defines observable triggers. When such evidence emerges, we envision a graduated response: minor concerns warrant design iteration (e.g., adjusting defaults to reduce dependency); serious concerns (e.g., longitudinal evidence of systematic deskilling) warrant escalation through the community review and open registry mechanisms in Section 4.3. This parallels clinical trial adverse-event reporting: structured escalation, not a single binary decision. We will add this response chain to Section 4.1.
>
> **Blur between research and deployment.** The paper draws a deliberate two-level distinction: the *thesis* — that ML research embeds normative assumptions — applies broadly, but the *practical mechanism* (Purpose Statements) is scoped to papers claiming real-world relevance (Section 4.1, Section 6). Basic capability research would not be the primary target of the Purpose Statement norm. Purpose evaluation is proposed as a new research direction for the field to invest in, not a universal review criterion. We will sharpen this language to make the two levels more immediately visible. Regarding the homework example in Section 3.3 specifically: a researcher building an educational model who optimizes for "correct answers" is making a normative choice at the research phase, one that risks student deskilling, before any deployment occurs. The proxy optimization begins when the objective is chosen, which is why this example applies to research, not only to deployed systems.
>
> **Deployment-context dependence.** This is a perceptive observation. We agree that purpose impacts are highly context-dependent. Purpose claims are meant to be conditional on deployment context, not global positive/negative verdicts on a method. The Purpose Statement's failure-mode and falsification components are designed to capture this kind of context-dependence.
>
> **Testability is hard.** The reviewer is right to press on this. The difficulty of testing purpose claims is real, and it is one reason Section 4.1 proposes a quality gradient rather than a single standard: a minimal Purpose Statement specifies beneficiaries and failure modes; stronger versions add falsifiable criteria. Even at the minimal level, stating failure modes creates accountability that currently does not exist. The framework is designed to be *generative*, surfacing research questions that current norms leave implicit — rather than imposing a uniform testing burden.

---

> > ### Author Rebuttal · Reviewer_JrsL · 2026-04-03
> >
> > Thanks for the rebuttal. Already pretty positive on this paper so going to keep my score.

---

### Official Review · Reviewer_TJAR · 2026-03-24

**Significance:** 2
**Argument Clarity:** 3
**Rating:** 4
**Confidence:** 4

**Questions:**

It would be helpful to cite earlier related work such as Barocas, Problem Formulation and Fairness.

**Alternative Views Section:**

Yes

**Compliance With Llm Reviewing Policy A Conservative:**

Affirmed.

**Discussion Potential:**

2

**Final Justification:**

I appreciate the rebuttal and the revisions it proposes. I have revised my score up accordingly.

**Paper Summary:**

The paper argues that machine learning research implicitly encodes normative assumptions through its choice of objectives, data, and metrics.   These assumptions should be made explicit through the lens of "human purpose", "understood as supporting people’s capacity to pursue meaningful, self-chosen life projects with agency." The authors propose community practices such as purpose articulation (structured statements specifying intended beneficiaries, mechanisms, and failure modes), purpose evaluation (measuring impacts on agency and meaning alongside task performance), and purpose governance (participatory processes for updating purpose frameworks). The paper extends existing documentation norms like model cards and datasheets to the realm of purpose.

**Position:**

Yes

**Position In Title:**

Yes

**Related Work:**

4

**Strengths And Weaknesses:**

Strengths:

The central underpinnings of the paper are strong: ML systems already embed normative assumptions through objective functions, metrics, and data choices.  Making those assumptions explicit is better than pretending they don't exist. The "purpose statement" is actionable and is something the ML community could reasonably adopt.

The paper does a good job presenting and engaging with possible weaknesses, objections, and opposing views. The paper clearly rebuts the idea that the work of purpose could be fulfilled by preferences, safety, or alignment.

Weaknesses:

The paper remains almost entirely at the level of "the community should." There is not yet empirical content, such as a case study showing what purpose governance would look like in practice. For a position paper this is acceptable, but it weakens the feasibility argument. It would be helpful to have one case study or example - even if an experiment is not possible, it would be great to work out a longer hypothetical case study for how this would go in practice.

The five research directions (Section 5) vary in specificity. Direction 2 (benchmarks for agency) has a concrete protocol sketch; Directions 3 and 4 are more abstract. "Institution-aware alignment" sounds valuable but the paper doesn't yet specify what a tractable research contribution in that space would be and how it would differ from e.g. a policy analysis.

The definition of "human purpose" in Section 2 covers a lot of philosophical ground quickly. Agency, meaning, and pluralism are each large concepts with contested boundaries. The paper does a good job defining terms, but doesn't always use the terms consistently with their definitions in this section.  The paper asserts compatibility across traditions (capabilities approach, self-determination theory, value pluralism) without fully grappling with tensions between them — e.g., Sen and Nussbaum themselves disagree about whether to specify a list of capabilities.

The relationship to existing documentation norms is underspecified. How should a Purpose Statement interact with a model card? Is it a section within one, a separate artifact, or a replacement?

The paper could better address power dynamics. Who writes the Purpose Statement matters — a Purpose Statement written by a lab about its own system is a very different accountability mechanism than one co-developed with affected communities. The paper acknowledges the issue of power, so it would be great to build on this analysis by saying who should write the purpose statement and how to ensure meaningful accountability in the creation of this document.

**Support:**

3

---

> ### Author Rebuttal · Authors · 2026-03-31
>
> We appreciate the reviewer's recognition that the central argument is strong and the Purpose Statement is actionable.
>
> **Lack of empirical content / case study.** Section 4.1 already provides an AI-assisted writing tool example with explicit failure modes (dependency, deskilling, homogenization) and falsification criteria. In the revision, we will extend this into a full end-to-end case study. To illustrate briefly here: the Purpose Statement identifies professional writers as beneficiaries and dependency as a key failure mode; evaluation tracks output diversity and self-reported creative agency longitudinally; if those measures worsen, governance mechanisms (open registry, community review from Section 4.3) trigger external review and design revision. We will develop this sketch into a complete worked example in the revised paper.
>
> **Research directions 3 and 4 lack specificity.** This is a fair point. To clarify what a tractable ML contribution looks like:
>
> - *Direction 3 (Pluralistic system behavior):* A concrete contribution would be a steering mechanism that lets users specify value-framework preferences (e.g., different cultural norms around directness in advice-giving, or varying preferences for autonomy vs. guidance in learning tools) and evaluates whether system outputs shift accordingly, extending the PRISM methodology (Kirk et al., 2024) from descriptive diversity measurement to controllable pluralism. This differs from simply training on diverse data; it requires explicit user-facing controls and evaluation of whether those controls are effective.
> - *Direction 4 (Institution-aware alignment):* A concrete contribution would be an evaluation benchmark that parameterizes institutional context: a suite of simulated deployment scenarios (e.g., the same AI tutoring system under school-aligned vs. engagement-optimizing incentive structures) against which ML researchers can test whether their system's purpose outcomes degrade. As reusable evaluation infrastructure, this is analogous to how robustness benchmarks parameterize distribution shift: an ML artifact that lets researchers stress-test systems against varied institutional conditions. We will sharpen this in the revision.
>
> **Philosophical tensions between traditions.** The reviewer raises a fair question about tensions between the capabilities approach, SDT, and value pluralism (e.g., Sen vs. Nussbaum on specifying capability lists). Section 2 already distinguishes our framework from the capabilities approach at the level of function: it is a community-wide research norm, not a philosophical account of the good life. Our framework operates at the level of *structural conditions* (capacity to choose, find meaning, differ) rather than *content* (which purposes to pursue), which deliberately brackets the Sen-Nussbaum disagreement rather than resolving it. We will make this pragmatic stance more explicit and audit the paper for terminological consistency.
>
> **Relationship to existing documentation norms.** Table 1 already distinguishes Purpose Statements from model cards: model cards document what a system *is*; Purpose Statements document what it is *for*. In practice, a Purpose Statement could function as a dedicated section within a model card or as a standalone artifact linked to one. We will make this practical relationship more explicit in the revision.
>
> **Power dynamics in Purpose Statement creation.** We acknowledge that a lab-authored Purpose Statement is an imperfect accountability mechanism. However, it is necessary if not sufficient: by forcing authors to write down their assumptions about beneficiaries, mechanisms, and failure modes, it creates a *falsifiable artifact* that external auditors and affected communities can subsequently critique. Without the statement, there is nothing concrete for the community to push back against. Section 4.3 already proposes participatory input mechanisms and open registries enabling exactly this external contestation. We will strengthen the revision by specifying a tiered approach: for research papers, authors draft the statement subject to peer review; for deployed systems, Purpose Statements should involve participatory input from affected stakeholders.
>
> **Citation of Barocas et al.** Thank you. We will add this citation on problem formulation encoding normative choices.

---

> > ### Author Rebuttal · Reviewer_TJAR · 2026-04-06
> >
> > Thanks for the thoughtful comments! I have revised my scores up.

---

### Decision · Program_Chairs · 2026-04-30

**Decision:**

Accept (regular)

**Comment:**

Reviewers felt the authors' position that ML research should be guided by an explicit model of human purpose was clearly and strongly articulated. Reviewers agreed that the authors did an excellent job of discussing weaknesses, objections and alternative views. In the authors' response, a key issue was clarified and emphasized regarding the purpose statement's testability and its difference from a product mission statement -- namely that the purpose statement is an explicit statement of failure modes intended to create accountability as it is subject to external verification. A number of strengths in the author's supporting arguments were also highlighted by reviewers including the pluralism aspect, the distinction from safety and the discussion of weaknesses and objections.

The paper is largely a call to action and while some of the call is actionable, the practical execution of other parts are unclear. Some reviewers wanted a larger case study to illustrate some concrete details of the call to action. As pointed out by a reviewer, research directions 3-5 are also more abstract in their directions. A reviewer also raised a concern about the lack of discussion about power dynamics regarding who writes the purpose statement; the paper would be strengthened with an inclusion of more details about this point.

Overall, this is a well-articulated position paper that presents a clear argument for a purpose model that would stimulate discussion of an important but often overlooked issue.